# Global, Regional, and National Burden of Road Injuries from 1990 to 2019

**DOI:** 10.3390/ijerph192416479

**Published:** 2022-12-08

**Authors:** Yifan Xu, Meikai Chen, Ruitong Yang, Muhemaiti Wumaierjiang, Shengli Huang

**Affiliations:** 1Department of Orthopedics, The Second Affiliated Hospital, Xi’an Jiaotong University, Xi’an 710049, China; 2Department of Intensive Care Unit, The Affiliated Drum Tower Hospital, Medical School, Nanjing University, Nanjing 210093, China

**Keywords:** road injuries, incidence, deaths, disability-adjusted life years, global burden of disease 2019

## Abstract

(1) Background: Understanding occurrence can help formulate effective preventative laws and regulations. However, the most recent global burden and road injuries (RIs) trends have not been reported. This study reports the burden of RIs globally from 1990 to 2019. (2) Methods: RIs data were downloaded from the Global Burden of Disease 2019. Incidence, deaths, and disability-adjusted life years (DALYs) described the trend and burden of RIs. We calculated age-standardized rates (ASRs) and estimated annual percentage change (EAPC) for the above indexes to evaluate the temporal trend of RIs. We evaluated the social-demographic index (SDI) with epidemiological RI parameters and reported proportions of age-standardized rates due to RI. (3) Results: In 2019, the global incidence of RIs reached 103.2 million. The EAPC of RI incidence increased, whereas deaths and DALYs decreased. Age-standardized incident rate (ASIR) was highest in low-middle SDI regions, age-standardized death rate (ASDR) was high in middle SDI regions, and age-standardized DALYs increased in low SDI regions. The highest accident rates were found in those aged 20–24 years old. Cyclist injuries were the leading RIs (34%), though pedestrian and motor vehicle accidents were the leading cause of death (37.4%, 37.6%) and DALYs (35.7%, 32.3%), respectively. (4) Conclusions: Over the past 30 years, RIs incidence increased annually, though death and DALY rates decreased. RIs places a considerable burden on public health in low SDI countries. Data should be used to develop and implement effective measures to reduce the burden of RIs.

## 1. Introduction

Recently, road safety has raised many concerns. The UN High-Level Meeting on Global Road Safety, held in 2022, reported the annual global burden of road injuries (RIs) to be 1.35 million deaths and 50 million physical injuries and disabilities [1,2,3]. In large-scale car accidents, the victims of RIs vary, including vehicle drivers, cyclists, pedestrians, and motorcyclists [4]. Road traffic accidents (RTAs) are the primary cause of death in people aged 5–29 years old [5]. In 75 developing countries, reports predict that by 2030 there will be 3 million deaths and 7.4 million injuries and disabilities following severe RTAs in people aged 10–24 years old [6]. Older age, female sex, and lower income status, combined with post-traumatic stress disorder, are factors associated with more severe RIs [7].

Moreover, lower limb RIs are associated with a lower quality of life [8]. RIs are divided into five subtypes: pedestrian injuries, cyclist injuries, motorcyclist injuries, motor vehicle injuries, and other road injuries. Furthermore, in some developing countries, neurotrauma caused by the five RI subtypes accounts for approximately 79% of all neurosurgical patients [9]. Moreover, the highest incidence for all five RIs subtypes is found in those aged 20–29 years old [9].

A global status report showed no significant decrease in RI death rates between 2010 and 2016 globally; however, RI death rates in African and low-income countries increased between 2010 and 2016 [10]. Notably, a systematic meta-analysis including 39 studies among 15 African countries showed that, during the 1990–2015 period, the RI incidence rate increased from 40.7 per 100,000 people to 92.9 per 100,000 people, and the RI death rate simultaneously decreased from 19.9 per 100,000 people to 9.3 per 100,000 people. However, the results vary, and it must be considered that some African countries are less developed than others [11,12]. In the World Health Organization European region, RIs caused 85,000 deaths in 2016 and were considered a leading cause of death among people aged 5–14 years old. The highest death rate in Tajikistan was equal to a seven-fold change in the RIs death rates of Switzerland and Norway, and this difference has continued to increase over recent years [13].

Aside from physical injury and disability, RIs cause a major economic burden, both in terms of cost and property loss. Financial losses due to RIs are huge, with total losses taking many factors into account, including property loss, in-patient treatment costs, rehabilitation expenses [14], post-trauma productivity/time loss, indirect time costs, legal costs, and public/private insurance costs [15]. Total economic costs vary among countries and territories with different social-economic indexes. In some countries with higher social-economic status, the cost of RIs accounts for a smaller proportion of the corresponding gross domestic product (GDP) than in low-GDP countries. For example, in Spain, RIs only cost 0.04% of the GDP [16]. In China, Tan et al. estimated the economic burden of RIs to be 72.6 billion USD for the whole country in 2017, accounting for 0.6% of the GDP for all nations [17]. In Iran, in 2013, the economic cost of RIs accounted for 6.64% of the gross national income (39 billion), far above the global average [18]. In undeveloped countries such as Nepal, the economic burden of RTAs in 2017 was around 122.88 million, equating to 1.52% of GDP [19]. In recent years, traffic rules and legislation implementation have improved conditions. In China, the revised road traffic safety law came into effect in 2011, and the number of years of life lost (YLLs) subsequently decreased significantly from 1133.14 in 2011 to 848.87 in 2019 [20] per 100,000 people.

Previous studies only demonstrated RIs conditions in a small number of regions and lacked consideration of the latest global conditions [10,21]. These studies included specific cities [22,23], countries and territories [19,24,25], continents [11], or regions [13,26]. Furthermore, these investigations were performed in the past and need to be reproduced. It is also important to consider the newest conditions. We wonder if tendencies in the different levels (world, regions, continents, and countries) have changed in the last three decades. Thus, this study aims to report the burden of RIs globally from 1990 to 2019. The Global Burden of Disease (GBD) database uses comprehensive and sophisticated analytical data on RIs, including the most recently known morbidity and mortality rates [27]. RIs have been studied extensively in various countries; however, it is also important to understand the global trend of RIs. In this study, we were granted permission to use the GBD 2019 online database to investigate the percentage of subgroup injuries in RIs. Subgroups were determined by searching the GBD 2019 database.

## 2. Materials and Methods

### 2.1. Overview and Definition

The GBD 2019 database (https://ghdx.healthdata.org/gbd-2019, accessed on 30 August 2022) systematically calculates the burden of 369 diseases and injuries and 87 behavioral, environmental, occupational, and metabolic risk factors among 204 countries and territories. In this investigation, we used the number and rate of RI incidence, deaths, disability-adjusted life years (DALYs), years lived with disability (YLDs), YLLs from 1990 to 2019 among men and women, social-demographic index (SDI) regions, and GBD regions; the percentage of the five RI subtypes is also presented for each parameter. This study aimed to provide a global summary of RIs to help determine and evaluate the global burden from 1990 to 2019. The methodology of the study has been previously elucidated [7,28]. Burden measures were normalized based on population and reported per 100,000 to remove the effect of age structure. According to the official guide and the previously selected parameters [9,29,30], we chose the number and rate of five RI metrics, including deaths, DALYs, incidence, YLLs, and YLDs. Premature mortality is measured by YLLs, which uses death frequency and age at death to estimate the years of potential life lost [29,31]. YLDs measures the impact and burden of a disease or illness on the affected person’s quality of life over both the short and long term before it either resolves or death occurs. DALYs is a time-based measure combining both YLLs and YLDs [32].

### 2.2. Socio-Demographic Index

The SDI describes socio-demographic development. The SDI fluctuates from 0 to 1.0 and is defined by method of income, average years of schooling, and fertility for each GBD location and year. Specifically, the SDI is calculated by the average educational attainment in members of the population aged >15 years, total fertility rate for those <25 years of age, and lag-distributed income per population. The 204 countries and regions are divided into five groups by SDI quintile values: high SDI (0.80–1.0), high-middle SDI (0.70–0.80), middle SDI (0.61–0.69), low-middle SDI (0.46–0.60), and low SDI (0–0.45) [33]. To determine the correlation between SDI index and EAPC, we used the SDI index values of 204 countries and regions from 1990 to 2019 shown on the official website (https://ghdx.healthdata.org/, accessed on 30 August 2022) (Appendix A).

### 2.3. Data Sources

The methodology of GBD 2019 research has been detailed in many other studies [31,34,35]. Firstly, we chose the GBD estimate by searching ‘CAUSES DEATH OR INJURY’. The most recent RI data for estimating the number and rate of deaths, incidence, and DALYs were subsequently downloaded from GBD 2019. GBD officials divided 204 countries and territories into 21 regions according to geographic and other factors [28]. The details of these regions are listed in the Appendix A.

### 2.4. Statistical Analyses

In this study, we assumed all the data including incidence, deaths, DALYs, YLDs, and YLLs were log-normally distributed. We applied the same method used in other papers in the GBD study design to calculate significance [30]. This study presents the 95% confidence intervals (CIs) calculated using 2.5th and 97.5th percentiles. ASRs are indicators that demonstrate adjustments for age structure among population size (number per 100,000 population). We calculated the estimated annual percentage change (EAPC) to determine the temporal trend in various age-standardized rates (ASR) of RIs; the detailed methods have been well established in previous research [28]. A linear regression model of ASXR was used to calculate the estimated annual percentage change (EAPC) [34,36]. Spearman’s correlation coefficients were used to determine the relationship between the ASXR and SDI. In the correlation analysis, the Pearson correlation coefficient was a positive number and the *p*-value was <0.05, demonstrating a significant positive correlation. All statistical analyses were performed using R statistical software, version 4.2.0. Adobe Illustrator 2021 was used to combine the images and statistical figures.

## 3. Results

### 3.1. Incidence of RIs Increased Globally from 1990 to 2019

Globally, the incidence of RIs increased 63.3% from 63.2 million [53.43–73.8] in 1990 to 103.2 million [86.9–121.3] in 2019 (Table 1). Of these RI cases, 35.6 million [30.2–41.7, 34.5%] were in women and 67.6 million [56.7–79.7, 65.5%] were in men (Table 1 and Figure 1A). The ASIR of RIs increased from 1,192.7 [1,017.76–1,389.27] per 100,000 people in 1990 to 1298.55 [1092.23–1529.42] per 100,000 people in 2019, with a 40% (26–55) increase calculated by EAPC from 1990 to 2019. In men, the ASIR in 2019 was 1697 [1,418.87–1,996.22] per 100,000 people, with an 11% (0–21%) increase calculated by EAPC from 1990 to 2019. In women, the ASIR in 2019 was 896.32 [759.59–1050.35] per 100,000 people, with a 57% [39% to 74%] increase calculated by EAPC from 1990 to 2019 (Table 1).

Focusing on the level of the five SDI regions (Table 1 and Figure 1A), the low-middle SDI region had the highest RI incidence among SDI regions in 1990, which was initially 15.0 [14.6–20.1] million but increased two-fold to 31.1 [25.8–37.0] million in 2019. However, the low SDI region, calculated by the absolute number, had the highest change rate of +132%. The RI incidence rate of the high SDI region significantly decreased by 16.7% from 8.34 [7.33–9.49] million in 1990 to 7.45 [6.43–8.61] million in 2019. The ASIR was significantly decreased in the high/high-middle SDI regions and increased in the other three SDI regions. Interestingly, the ASIR of low-middle SDI regions (1,443.53 [1,216.45–1,694.36]) was marginally lower than high-middle SDI regions (1,451.94 [1,237.87–1,693.16]) in 1990; however, in 2019, the ASIR for these regions was 1,752.61 [1,464.86–2,079.37] and 739.88 [629.42–865.09], respectively. The EPAC in these SDI regions showed that regions with higher SDI decreased more significantly (−1.37 [−1.47 to −1.27]), whilst regions with middle SDI increased more significantly (0.99 [0.83 to 1.15]).

Among GBD regions (Table 1 and Appendix A), the absolute incidence of RI increased, with the highest incidence observed in 2019 in South Asia (43,565,410 [35,693,282–52,454,188]) and the lowest observed in Oceania (85,961 [72,926–99,393]); meanwhile, absolute incidence in western sub-Saharan Africa significantly increased from 1,938,780 [1,659,937–2,244,320] to 5,100,399 [4,308,008–5,941,026]). Moreover, incidence in the high-income Asia Pacific region significantly decreased from 1,544,957 [1,346,761–1,785,737] to 890,891 [764,685–1,036,965]. In 1990, the ASIR fluctuated from the lowest values in East Asia, 535.26 [450.72–630.33], to the highest values in Eastern Europe, 3,472.2 [2,884.07–4,151.44]. However, the ASIR for Eastern Europe still ranked first (2,600.36 [2,123.05–3,142.83]) and the high-income Asia Pacific region showed the lowest ASIR in 2019 (473.34 [397.64–566.77]). From 1990 to 2019, the EAPC in these GBD regions showed that the high-income Asia Pacific region primarily decreased (−2.53 [−2.71 to −2.35]) and western sub-Saharan Africa increased rapidly (Table 1).

Next, we focus on the level of countries and territories (Figure 2 and Appendix A). India showed the highest incident number both in 1900 (16.2 [13.2–19.6] million) and in 2019 (36.6 [29.8–44.3] million) (Figure 2A), an increase of approximately 126% (Figure 2C). Portugal had the lowest incidence number in 2019 (376.37 [324.12–439.02]) and had the lowest EAPC (−3.88 [−4.14 to −3.63]) (Figure 2B). The ASIR of the Russian Federation was also high both in 1990 (3,788.05 [3,096.01–4,597.72]) and in 2019 (2,801.2 [2,262.62–3,411.69]), but EAPC decreased by −0.72 [−0.94 to −0.49].

### 3.2. Death of RIs Decreased from 1990 to 2019 Globally

The number of deaths caused by RIs had an upward trend over the past three decades, from 1,113,411 [1,047,007–1,209,611] in 1990 to 1,198,289 [1,060,041–1,304,831] in 2019 (Table 2 and Figure 1B). The number of deaths increased in men but decreased in women (Figure 1B). ASDR changed globally from 21.92 [20.65–23.86] in 1990 to 14.99 [13.29–16.32] in 2019 per 100,000, with an EAPC of −1.29. In contrast to the number of deaths, the ASDR in men and women was 22.79 [19.37–25.11] and 7.36 [6.7–8.06] in 2019, respectively, with an EAPC of −1.14 [−1.29 to −0.98] and −1.73 [−1.88 to −1.58] (Table 2).

At the regional SDI level (Table 2 and Figure 1B), the middle SDI region had the most extensive case numbers both in 1990 (412,039 [383,879–459,016]) and in 2019 (438,076 [382,035–480,744]). The low SDI region had the lowest case numbers in 1990 (109,689 [91,807–127,498]). However, in 2019, the high SDI region was ranked lowest (109,511 [101,734–118,418]). The ASDR of the high SDI region was lowest in both 1990 (17.37 [17.04–17.7]) and 2019 (9.26 [8.59–10.08]), and also had the most negative EAPC (−2.47 [−2.61 to −2.33]) among the other four SDI regions. The ASDR in the middle SDI region ranked first in 1990 (26.25 [24.42–29.36]) but went down to second in 2019 (17.45 [15.32–19.13]). As can be seen in the curve, similar to the global decrease, ASDR in all five SDI regions decreased over time (Table 2).

For GBD regions (Table 2 and Appendix A), the number of deaths was highest in East Asia in 1990 (244,917 [210,138–333,881]) and 2019 (262,312 [217,269–306,437]). Oceania had the lowest death number both in 1990 (1,505 [1,217–1,862]) and in 2019 (2,814 [2,179–3,703]), though the rate of increase was greater than other GBD regions (−0.40 [−0.45 to −0.35]). Central sub-Saharan Africa was top in terms of age-standardized DALYs both in 1990 (54.83 [42.8–70.26]) and in 2019 (41.92 [30.06–58.49]). The lowest ASIRs were observed in southern Latin America in 1990 (14.48 [14.12–14.82]) and the high-income Asia Pacific region in 2019 (4.82 [4.49–5.67]). The high-income Asia Pacific region also had the lowest EAPC (−5.19 [−5.48 to −4.9]) (Table 2).

At the countries or territories level (Appendix A), China had the highest number of deaths both in 1990 (230,499 [195,994–318,523]) and in 2019 (250025 [205,596–294,524]). The lowest number of fatalities were observed in Palau, Nauru, Tuvalu, San Marino, and Monaco in 1990, with an average of three, and in Greenland and Monaco in 2019, with an average of two. In 1990, Oman (109.33 [85.67–134.85]) and Jamaica (5.18 [4.88–5.56]) held the highest and lowest positions when ordered by ASDR (Appendix A); in 2019, the Central African Republic was highest (67.12 [48.33–90.71]) and Singapore was lowest (2.64 [2.39–2.93]). Most countries have a negative EAPC value, though the EAPC value for Mexico increased (EAPC: 3.9 [2.29 to 5.54]) (Appendix A).

### 3.3. Age-Standardized DALYs attributable to RIs Decreased from 1990 to 2019 Globally

At a global level, the DALYs attributable to RIs increased from 71,212,240 [66,408,720–77,034,750] in 1990 to 72,901,326 [64,830,881–80,193,702] in 2019 (Table 3 and Figure 1C). The number of DALYs between women and men has not significantly changed over the past three decades. Age-standardized DALYs decreased gradually from 1,329.47 [1,235.48–1,435.89] in 1990 to 917.94 [814.15–1,011.37] per 100,000 population, with an EAPC value of −1.26 [−1.4 to −1.13]. A decrease was observed in men and women, as with the global EAPC, with values of −1.69 [−1.81 to −1.57] and −1.09 [−1.24 to −0.94], respectively.

In the different SDI index locations (Table 3 and Figure 1C), the middle SDI region had the highest number both in 1990 (25,957,977 [24,113,339–28,315,794]) and 2019 (24,862,198 [22,069,837–27,303,133]). In 1990, the lowest number of DALYs was observed in the low SDI region, whereas the high SDI region had the lowest DALYs in 2019. Age-standardized DALYs in the high SDI region were the lowest for 1990 (1,059.2 [1,011.15–1,111.18]) and 2019 (562.11 [515.83–617.59]), with a rapidly decreased EAPC of −2.48 [−2.61 to −2.34] (Table 3).

Regionally (Table 3 and Appendix A), DALYs were observed to be the highest in East Asia in 1990 (145,289.57 [126,698–192,614.2]) and in South Asia in 2019 (18,232,475 [15,045,667–21,179,151]). DALYs were the lowest in Oceania in 1990 (92,990 [75,929–115,406]) and Australasia in 2019 (115,650 [105,218–127,570]). We observed that central sub-Saharan Africa possessed the highest number of age-standardized DALYs in 1990 (3,085.19 [2,423.86–3,728.34]) and 2019 (2,190.04 [1,597.63–2,895.56]), with a decreased EAPC of −1.14 [−1.3 to −0.98]. The high-income Asia Pacific region ranked lowest in age-standardized DALYs in 2019 (276.35 [249.83–319.72]) and had the largest decrease in EAPC (−5.16 [−5.45 to −4.87]) (Table 3).

Nationally (Appendix A), Tokelau had the lowest DALYs in both 1990 (16 [12,13,14,15,16,17,18,19,20,21,22]) and 2019 (10 [7,8,9,10,11,12,13,14]). In 1990, China had the highest DALYs (13,739,515 [11,907,510–18,430,633]). However, the condition changed slightly after three decades, with India replacing China as the country with the highest DALYs (15,593,133 [12,585,561–18,249,333]). Oman had higher age-standardized DALYs in 1990 (5,317.31 [4,261.11–6,476.16]) than the other nations. In 2019, the Central African Republic was highest (3,543.87 [2,559.38–4,773.75]). Jamaica had the lowest age-standardized DALYs in 1990 (336.46 [303.56–374.06]), though Singapore (189.84 [166.62–216.97]) had the lowest age-standardized DALYs by the end of 2019 (Appendix A). As depicted in the Appendix A, the fastest growth in age-standardized DALYs was observed in Lesotho (2.37 [1.94 to 2.79]), whereas the fastest decrease was seen in Portugal (EAPC: −6.52 [−6.91 to −6.14]) (Appendix A).

### 3.4. Correlation between SDI Index and EAPC

Overall, age-standardized incidence rate does not correlate with the tendency index (EAPC) globally (R = 0.3, *p* > 0.05). For regions with different SDI indexes (Figure 3A,B), countries with low SDI indexes have a negative relation to the EAPC (ASIR) (R = 0.54, *p* < 0.05). In research on the association between the EAPC and ASDR/SDI (Figure 3C,D), the results are similar to RI incidence; there was no relationship between the 2019 age-standardized death rate and the EAPC from 1990 to 2019 (R = 0.43, *p* > 0.05). However, in countries with a higher SDI index, the EAPC rate decreases. Notably, there is a positive relationship in the regression model between EAPC and age-standardized DALYs (Figure 3E,F). Countries with higher age-standardized DALYs have a higher EAPC (R = 0.55, *p* < 0.05). Evaluation of the correlation between SDI and EAPC showed high SDI regions or countries have stronger negative correlation rates (R = 0.58, *p* < 0.05).

### 3.5. The Features of RIs in Different Age Groups

We explored the age-related population metric in different genders. In 2019, as shown in Figure 4A, the incidence of RIs was mainly observed among people aged 20–24 years old. This was the same for both women and men. Over the years, a decreasing RIs tendency was seen in men, whilst the incidence rate remained high in women except in women aged over 60, where RIs tendency decreased. Absolute RIs numbers were also lower in women than in men. The death rate (Figure 4B) presents a similar curve that decreases alongside age increases (though not for people younger than 19 years old). There is a smooth curve for those aged 20–50 years old. Similarly, the death rate in women is lower than in men. Interestingly, the DALYs rate (Figure 4C) in both genders is different to incidence and death rate. In women, the DALYs rate peaks in the 70–74 age group rather than in the 20–24 age group with the maximum incidence rate. In men, this is different; the peak rate occurs in the 20–24 age group, though with increasing age the DALYs rate steadily declines. However, there is a considerable gap between the genders. Furthermore, when calculating the DALYs rate in the past 30 years, excluding gender differences, people < 19 years old show different conditions. Here, results showed that the difference between 1990 and 2019 in the 20–24 age group was the largest among all age groups in terms of calculated incidence rate (Figure 4D), and the death rate was lower in 2019 than in 1990 in all age groups. This suggests that, with increasing age, the difference in death rate between 1990 and 2019 became more significant (Figure 4E). When comparing the different DALYs rates, a large difference is seen in the 20–24 age group (Figure 4F).

### 3.6. The Proportion of Five Different Subgroups of Road Injuries

Observing the incidence of RIs globally, cyclist road injuries (34%) and motor vehicle road injuries (24.8%) account for the majority of RIs. In the high SDI region, the proportion of motor vehicle injuries increased from 19.6% to 46.2% in 2019; however, cyclist injuries decreased from 40.9% to 25.4% (Figure 5A).

The other four subtypes of injuries showed no apparent difference among the five SDI regions. In GBD regions, the pattern varies. Pedestrian road injuries account for no more than 30% of RIs. In three sub-Saharan African areas, pedestrian injuries rank in the top three among the five injury types. Motorcyclist accidents among GBD regions show a significant difference, with tropical Latin America and Southeast Asia accounting for 35.6% and 37.6%, respectively, making motorcyclist-related RIs the most common in those regions. Motor vehicle injuries make up the majority of RIs in most areas, accounting for over 50% in some regions (Australasia: 51.2%; Eastern Europe: 51.2%; high-income Asia Pacific: 59.7%). In contrast, motor vehicle-related RIs account for just 10.1% of RIs in tropical Latin America.

Focusing on deaths caused by RIs (Figure 5B), the global death rate for pedestrians in terms of RIs (37.6%) is equal to that of motor vehicles (37.4%). However, the cyclist accident group only accounts for 5.4% of all RIs deaths, even lower than the motorcyclist accident group. Among the five SDI regions, the deaths vary. In the highest SDI index region, motor vehicle accident RIs are the highest cause of death (58.9%); this proportion is also highest when compared to the other four regions in terms of motor vehicle accident RIs. We also focused on the GBD areas related to motor vehicle accidents and found that the tendencies in high-income North America and the high-income Asia Pacific region were contradictory. In the former region, this subtype leads to 69.7% of deaths, while in the latter region they only account for 31.7% of deaths. Motorcyclist accidents influence the death percentage differently, similar to tropical Latin America (34.4%) and Southeast Asia (34%) where this subtype accounts for a large number of deaths. Interestingly, though the incidence of cyclist RIs is high in many GBD regions, the rate of deaths caused by cyclist RIs ranks bottom among the five injury subtypes.

Finally, the indication of DALYs shows a similar pattern to deaths caused by RIs (Figure 5C). Globally, the descending order of DALYs in terms of subtype is motor vehicle (35.7%), pedestrian (32.3%), and motorcyclist (21.9%). In high SDI regions, the majority of DALYs are caused by motor vehicle RIs (58.2%), and this trend is also seen in low SDI regions (42%). Pedestrian injuries cause the highest rate of DALYs in low SDI regions (36.2%), but the lowest rate in high SDI regions (18.8%). The lowest DALYs caused by motorcyclist injuries occur in low SDI regions, while the number is higher in low-middle SDI regions. In GBD regions, southern sub-Saharan Africa (61.5%), high-income North America (65.7%), and Australasia (60.9%) are the top three in terms of DALYs owing to motor vehicle RIs. Interestingly, motor vehicle RIs in all regions occur at a rate of over 30%. In East Asia and Andean Latin America, 48.5% and 40.5% of DALYs are attributed to pedestrian road injuries. This trend decreases in Australasia (14.6%) and high-income North America (15.7%). DALYs caused by motorcyclist crash-related RIs differ, with this subtype accounting for 40.8% in Southeast Asia, 39.6% in tropical Latin America, and just 4.3% in southern sub-Saharan Africa.

## 4. Discussion

RIs can severely impact personnel and society [4]. Therefore, in this study, we describe the global burden of RIs in different regions, including quinquepartite SDI regions, GBD regions, and countries/territories from the GBD 2019 database. The results show that the incidence of RIs did not continually increase during the 2000–2005 period (where the global incidence rate is at a plateau). After 2010, RI incidence declined annually; therefore, this period is viewed as the ‘Decade of Action for Road Safety‘ [2]. There may be various reasons that can account for this phenomenon, including global road improvements [22,26,36,37,38], related legislation implementation [13,22,24,39], and the state of war [40]. The absolute number of RI incidences was highest in India (low-middle SDI region), and the highest numeric RIs value was observed in South Asia. This point has been verified in many previous studies [25,26,34]. After age normalization, the condition varies. India, Pakistan, and Bangladesh belong to the low-middle SDI region, and South Asia has the highest incidence number; however, Russia (high-middle SDI) ranks top in terms of calculated ASIR. The EAPC in Portugal and all high-income Asia Pacific countries (high SDI region) mostly decreased, indicating an increasing reduction in RIs incidence. Moreover, this suggests that effective measures should be implemented to prevent road accidents [13]. We also observed that the RIs incidence rate in men was continuously higher than in women, especially in the 20–24 age group. As traffic laws and medical care improved, the death rate in all regions decreased annually, with an EAPC of −1.29. Up to 2019, the highest two absolute death numbers were observed in China (middle SDI) and India (low-middle SDI). After age normalization, the Central Africa Republic in central sub-Saharan Africa had the highest ASDR, and Singapore in the high-income Asia Pacific region had the lowest ASDR. This suggests that some African regions still need to implement measures to prevent deaths caused by RIs [11,26]. Regarding differences in gender for deaths caused by RIs, similar to what has been observed in previous incidence research, the RIs death rates in men were higher than in women. A potential explanation for this may be that, with increasing age, the physical condition of men declines more rapidly than women [39,41]. Additionally, some economic and social factors should be considered, such as the exposure level restricted by regional legislations [42] and levels of occupational-related mobility [43]. It is also worth noting that a higher amount of driving-related worry and a more robust level of self-control have been postulated to explain women’s underexposure to driving [44,45].

The change in DALYs indicated by the results demonstrates that diseases burden public health in across regions. DALYs refer to disability-adjusted life years, which integrates the years of life lost and the years lived with disability [15,16,28]. Among all the SDI regions, the DALYs rate continually decreased. The low SDI region had a higher DALYs rate than the other four regions, especially after 2010, in which most middle SDI regions issued related laws to prevent RIs. Low SDI, low-middle SDI, and middle SDI regions present similar AS-DALYs during the 1990–2010 period in total and/or in men. Conversely, in women, the DALYs rate remains higher in low SDI regions than in other regions. In 2019, the highest DALYs absolute value was observed in India (South Asia, low-middle SDI). After age normalization, the Central African Republic (central sub-Saharan Africa, low SDI) had the highest DALYs rate, indicative of a serious situation in some African countries. In some developed countries such as Singapore (high-income Asia Pacific, high SDI), the AS-DALYs rate is only one-eightieth of the rate in the Central African Republic. Results show that some of the measures taken in some South Asian countries and high-middle SDI regions are making great progress in reducing the burden of RIs with an associated EAPC of −2.4, which is a three-fold improvement when compared to low-middle SDI regions (−0.59).

The results suggest low-middle SDI countries are in an awkward position; on the one hand, they need to build more roads to promote development, and on the other hand, increasing accident burden hampers economic growth. Though their death rate and DALYs rate are even lower than the low SDI region, low-middle SDI countries (such as India) have the highest number and age-normalization rate when calculated by incidence. In India, the RIs incidence rate is not low, owing to India having the second-largest population in the world. The situation is difficult for most low-middle SDI countries; though the speed of increase in RIs incidence is lower (ASIR-EAPC), the rate of decreased deaths (ASDR-EAPC) and DALYs (AS-DALYs-EAPC) remains lower than in the other four SDI regions. This phenomenon appears obviously in South Asian countries [26]. Low SDI countries have a higher ASDR and number of AS-DALYs compared to low-middle SDI countries. We should consider that the medical level in some low SDI regions still represents a public health problem, which has been observed in some African countries [3,11]. Thus, in most low SDI countries, the burden of RIs is still huge. Due to economic development and road conditions in less developed countries, the overall speed of various vehicles may be slower than those in more developed countries, such as middle and high-middle SDI regions (e.g., China, Russia). Hence, when traffic accidents occur, the accidents in middle SDI regions are more severe, causing more deaths and DALYs than accidents in low-middle SDI countries [25,34]. For some middle SDI countries, though the government can afford the economic loss of RIs, the EAPC in ASIR is still higher than even some low-middle SDI countries. Taken together, the conditions in low-middle SDI regions suggest a cautious approach to the prevention of RIs; for those in low-middle regions that develop towards a higher SDI index, low-middle SDI regions could be an example when balancing economic considerations and road injury prevention [7].

Furthermore, whilst non-adults (aged < 19) are not allowed to drive a vehicle in most regions, they can physically move or chase one another more casually [5,46]. In our study, we conclude that the incidence of RIs increases with increasing age. However, the number of deaths related to RIs is low for those under the age of 14. However, there is a sharp shift in the curve of fatalities for those in the 15–19 age group. A study from China shows that RIs were one of the most common causes of death in children aged 5–19 years old and ranked just after neoplasms [47,48]. This can be explained by the fact that legal guardians take care of their children, especially those under 14. However, adolescents change physically and psychologically, including stages of teenage rebellion, translation to maturity while lacking social knowledge, exposure to alcohol, cocaine, and other psychotropic drugs, violent tendencies, and a lack of guardian supervision, all of which increase adolescents’ risk of RIs. As such, many studies appeal for the implementation of legislation to protect the safety of children and adolescents [5,49]. Based on our study findings, current legislation has been successful. Compared to 1990, the incidence of RIs for those aged <14 years old was even lower in 2019 than in past decades. Though incidence in the 15–19 age group remained higher than in 1990, the increase in rate is lower than in the adult group. Apart from the decline in incidence, the death rate and DALYs rate were all lower than in other age groups at all times. These data show the effectiveness of these legislations. Moreover, among the elderly, though the incidence rate among older adults negatively relates to age, the death rate/number maintains a high level. This phenomenon can be easily explained by the aged group being more vulnerable to external violence injuries owing to comorbidities and aging organs [39,41].

In general, RIs include five subtypes of injuries. We observe that different regions have different RIs subtype features. In Australasia, Eastern Europe, and high-income North America (countries and territories with a high SDI index), motor vehicle RIs occur more frequently. In tropical Latin America and South Asia, motor vehicle accidents occur the least. Notably, among all the regions, cyclist accidents happen most often. Though the incidence of cyclist accidents is high in most countries, the associated death rate and DALYs rate for this subtype in these regions account for only a small percentage of each metric. Taking eastern sub-Saharan Africa as an example, the incidence rate proportion reaches 50%, yet deaths and DALYs associated with this subtype only account for up to 5.6% and 9.9%, respectively, of the total metric. This abnormality can be explained by the following. Firstly, cyclists typically travel shorter distances and are equipped with fewer protections, which can explain the high incidence rates. Secondly, bicycles cannot travel with the same high velocity as motor vehicles, and this can help limit the severity of a collision. Thirdly, in countries and territories with more extreme climates and environments (such as tropical Latin America), there are fewer motor vehicles on the roads, reducing the severity of collisions between cyclists and motor vehicles [4,50].

Finally, the second subtype of injuries to address is motor vehicle accident RIs. Motor vehicles derive from several industrial revolutions. The number of vehicles in certain regions depends on the natural environment and economic development [13]. The incidence of RIs, deaths related to RIs, and resulting DALYs in high SDI regions or some developed countries remain at a high level. This has caused many governments to enact road laws to decrease the burden on entities [51,52]. Some accidents occur among different transportation forms, such as vehicle-pedestrian accidents [49,50,52] and vehicle–cyclist accidents [1,6,50]. In such accidents, pedestrians are more vulnerable to loss of life and disabling injuries, especially in older adults and children [5,7,23,39,49]. Our analysis results agree with these existing reports. East Asia and Andean Latin America have the most severe RIs burden on pedestrians, with deaths/DALYs exceeding 50% deaths. Hence, some laws aim to not only reduce the incidence of motor accidents but also provide a level of protection for pedestrians [20,49,52]. Therefore, regardless of the world group division, RIs differ. In developed countries, more pedestrian–motor vehicle injuries cause more deaths. In undeveloped countries, a higher absolute number of vehicles typically increases RIs incidence and death among pedestrians. Road injury is a multifield issue. The use of helmets and seat belts [53], drink driving bans [54,55], speed limitations [56], and restrictions on children’s usage of transport have been proven to reduce mortality by 20–40% [2]. Such approaches are critical to low- and middle-income nations, which account for more than 90% of annual road traffic fatalities and incidents [2,3]. As previously reported, certain African countries without helmet-related laws should implement helmet use policies to reduce RIs related to motorcycle crashes [53]. We recommend that legislation in low SDI regions and low-middle SDI regions be further developed around these four elements. Another solution is to improve road conditions by strengthening them, through implementation of cycle tracks [57,58] or through the addition of adequate street lighting [59]. It should be highlighted that the illumination element is still debatable and will need to be verified further in future investigations [59,60].

This investigation also has some limitations. The data from the GBD database were calculated using estimation and mathematical modeling, which can be affected by adjusted and confounding factors. Additionally, different healthcare diagnosis systems in various countries can provide a mistaken impression of incidence and death rate in the region through overdiagnosis and mistaken diagnosis. We also describe global cohort changes in this study but lack gender stratification studies. Hence, further research will investigate gender-related changes in RIs. Some risk factors that cause RIs are not included in this study. In a further study, we could explore the risk factors associated with RIs globally.

## 5. Conclusions

RIs remain a concern in many countries and territories. Globally, incidence decreased during the 2010–2015 period, after which the incidence increased to 103.2 million [86.9–121.3] in 2019. Low-middle SDI countries showed higher incidence levels (31.1 [25.8–37.0] million) in 2019; at the same time, low SDI countries had more RIs-related age-standardized deaths (19.9 [16.76–23.22]) and higher DALYs rates (1152.15 [977.14–1331.52]). The higher DALYs rate increases the burden on low SDI regions, affecting economic development. In high SDI regions, RIs incidence (−1.37 [−1.47 to −1.27]), death rate (−2.47 [−2.61 to −2.33]), and DALYs (−2.48 [−2.61 to −2.34]) decreased obviously, with higher EAPC values than other regions. For GBD regions, there is still a considerable burden on East Asia, South Asia, and sub-Saharan Africa in terms of RIs incidence rates, death rates, and DALYs rate, with a lower EAPC in the decreased indicators. High-income countries suffer less RIs burden, calculated by incidence, deaths, and DALYs, and can reduce the influence of RIs more quickly. Results show that men are more vulnerable to RIs, whatever the economic condition. People aged 20–24 are the leading victim group, though deaths are not highest in this group, owing to their robust recovery. Men over 20 years old have less RIs and DALYs with increasing age, though incidence and the DALYs rate remain high with increasing age in women.

Given both economic and environmental differences, the subtypes of RIs manifest differently in each region. Even though cycling injuries are most common worldwide, they are associated with fewer casualties and DALYs. Countries with a higher incidence of motor vehicles injuries, such as in Australasia, will likely see an increase in the proportion of motor accident deaths and DALYs. Tropical Latin America has more motorcyclist accidents. Consequently, the number of motorcyclist fatalities has become a significant percentage of all RIs. However, pedestrian RIs remain an enormous burden on governments. As such, we recommend that administrators pay more attention to these issues and develop regulations that protect pedestrian safety. Regardless of the type of RIs, we appeal to the use of helmets and seat belts, restrictions on children driving, and the prohibition of alcohol consumption.

## Figures and Tables

**Figure 1 ijerph-19-16479-f001:**
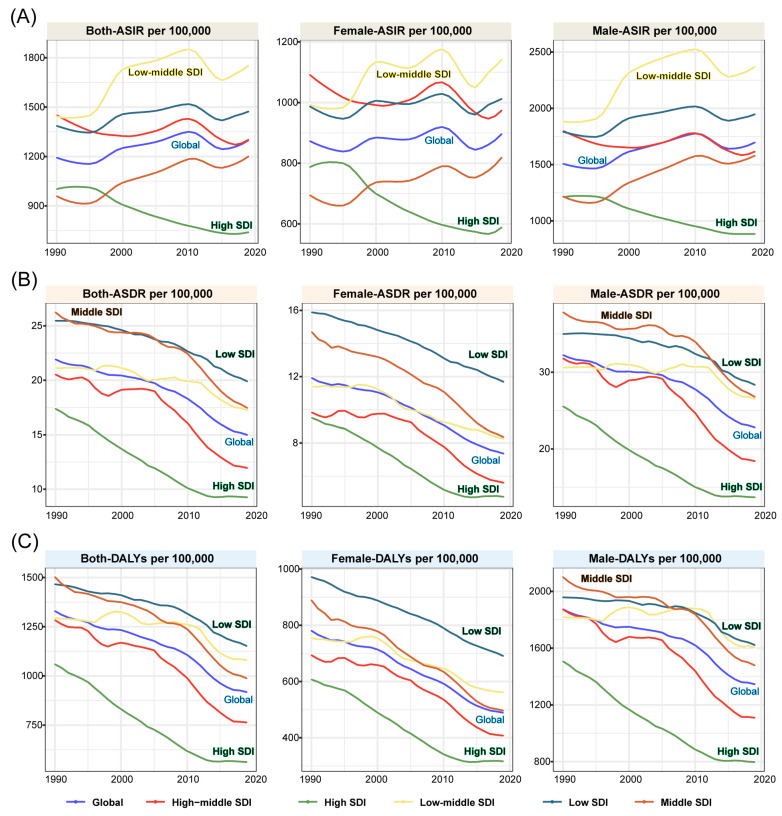
The burden caused by RIs in five SDI regions and globally from 1990 to 2019 in all populations, women, and men and calculated by age-standardized incidence (**A**), deaths (**B**), and DALYs (**C**).

**Figure 2 ijerph-19-16479-f002:**
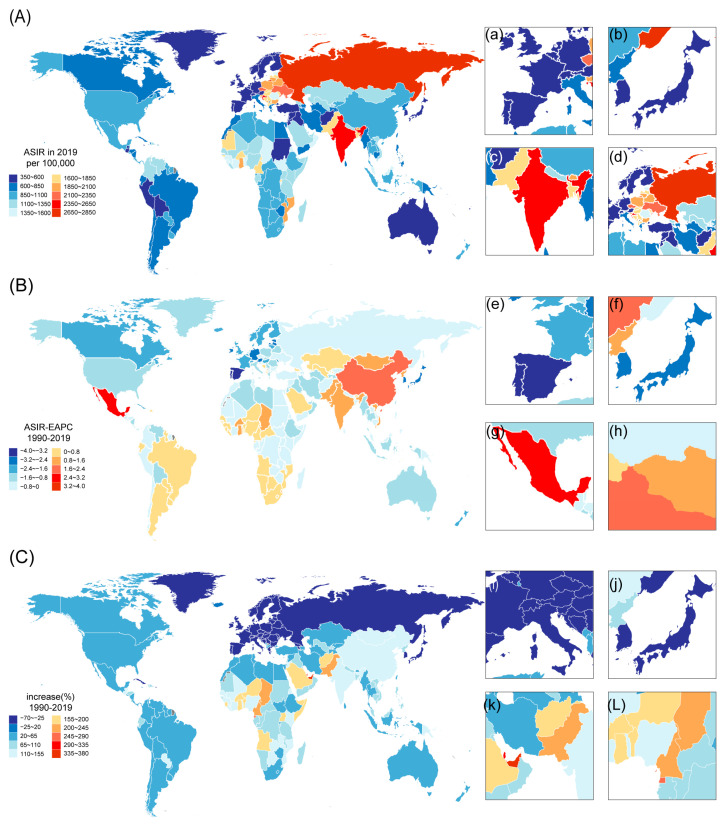
Incidence and its tendency from 1990 to 2019 among 204 countries and territories. (**A**) Age-standardized incidence rate (ASIR) in 2019. (**B**) EAPC of ASIR. (**C**) The absolute number for rate of incidence increase (%). Western Europe (**a**) and the high-income Asia Pacific region (**b**) with lower ASIR; South Asia (**c**) and Eastern Europe (**d**) with a higher ASIR. Western Europe (**e**) and the high-income Asia Pacific region (**f**) with lower ASIR-EAPC; central Latin America (**g**) and East Asia (**h**) with higher ASIR-EAPC. Most of Western Europe (**i**) and some of the high-income Asia Pacific region (**j**) with a lower increase in incidence; North Africa and the Middle East (**k**) and central sub-Saharan Africa (**L**) with a higher increase in incidence.

**Figure 3 ijerph-19-16479-f003:**
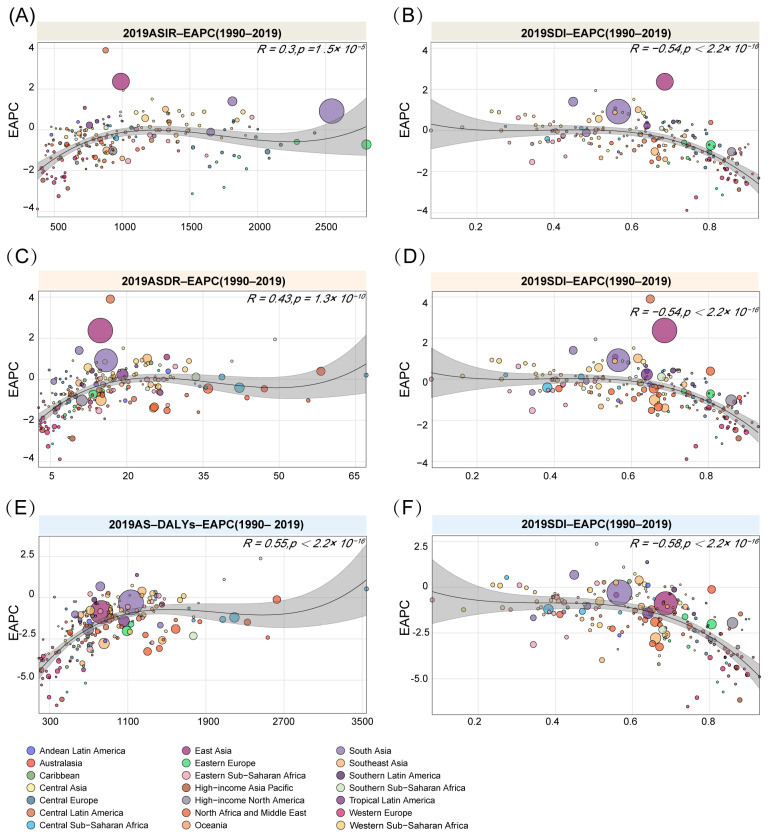
The relation between EAPC and ASXR/SDI index. (**A**,**B**) ASIR-EAPC correlation analysis with ASIR and SDI index; (**C**,**D**) ASDR-EAPC correlation analysis with ASDR and SDI index; (**E**,**F**) AS-DALYs-EAPC correlation analysis with AS-DALYs and SDI index.

**Figure 4 ijerph-19-16479-f004:**
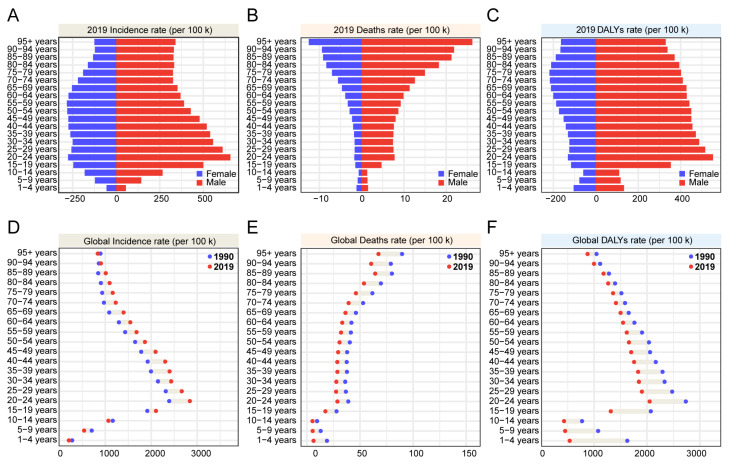
Incidence of RIs, deaths caused by RIs, and DALYs in different age groups. (**A**–**C**) The rates for various age groups in 2019 for both men and women; (**D**–**F**) the difference between 1990 and 2019 for the different age groups.

**Figure 5 ijerph-19-16479-f005:**
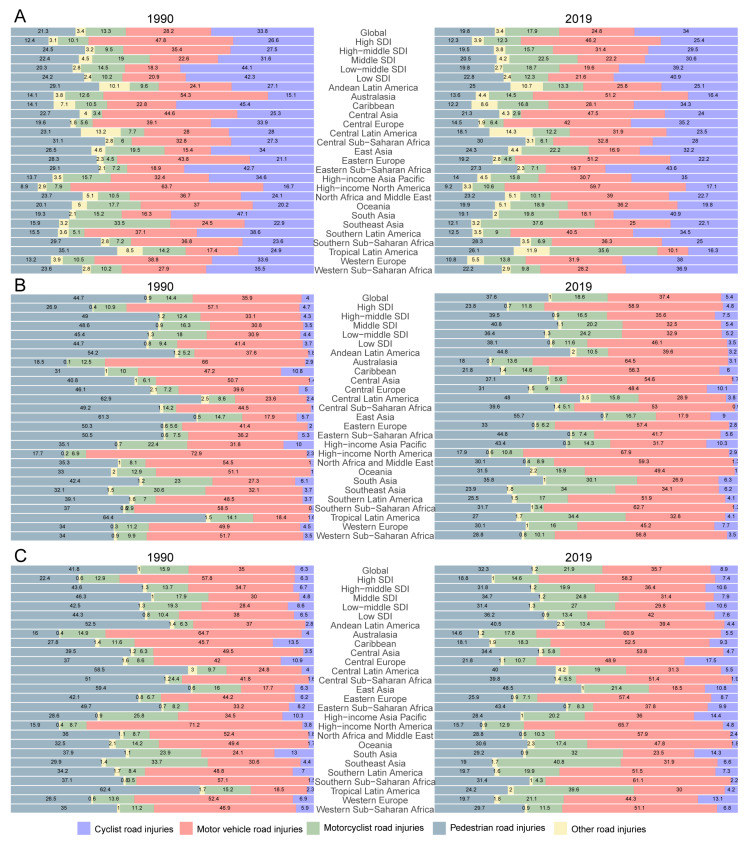
RIs subtypes in the different SDI regions and GBD regions. They are presented by incidence (**A**), deaths (**B**), and DALYs (**C**) for 1990 and 2019.

**Table 1 ijerph-19-16479-t001:** Incidence case, ASIR, and temporal trends of RIs in 1990 and 2019.

	1990		2019		1990–2019
	Incidence Cases No. ×10^2^(95%UI)	ASIR per 100,000 No. (95%UI)	Incidence Cases No. ×10^2^(95%UI)	ASIR per 100,000 No. (95%UI)	EAPC No. (95%CI)
Global	632,111.48 [534,279.84–738,477.05]	1,192.7 [1,017.76–1,389.27]	1,032,196 [868,741.7–1,212,732.78]	1,298.55 [1,092.23–1,529.42]	0.4 [0.26 to 0.55]
**Gender**					
Female	227,723 [193,850.64–266,947.98]	872.97 [750.27–1,016.94]	355,983.71 [302,250.53–417,143.88]	896.32 [759.59–1,050.35]	0.11 [0 to 0.21]
Male	404,388.49 [338,821.93–472,587.08]	1,507.91 [1,285.08–1,755.44]	676,212.28 [567,121.62–796,660.96]	1,697 [1,418.87–1,996.22]	0.57 [0.39 to 0.74]
**SDI region**					
High SDI	83,420.3 [73,261.69–94,943.33]	1,002.16 [879.75–1,145.59]	74,520.51 [64,333.9–86,089.78]	739.88 [629.42–865.09]	−1.37 [−1.47 to −1.27]
High-middle SDI	172,153.71 [145,735.49–201,179.03]	1,451.94 [1,237.87–1,693.16]	198,445.66 [168,498.82–233,544.11]	1,301.8 [1,095.24–1,533.14]	−0.19 [−0.33 to −0.05]
Middle SDI	163,427.1 [136,804.4–192,338.55]	958.7 [812.76–1,115.91]	302,824.2 [256,388.68–355,815]	1,200.4 [1,020.29–1,408.07]	0.99 [0.83 to 1.15]
Low-middle SDI	150,368.13 [124,535.41–178,961.84]	1,443.53 [1,216.45–1,694.36]	311,099.29 [258,062.91–370,624.82]	1,752.61 [1,464.86–2,079.37]	0.73 [0.46 to 1.01]
Low SDI	62,463.85 [52,539.49–73,608.15]	1,386.14 [1,191.35–1,604.27]	144,936.04 [120,963.11–171,604]	1,473.35 [1,263.76–1,711.09]	0.29 [0.16 to 0.41]
**GBD region**					
Andean Latin America	2,142.89 [1,842.92–2,490.35]	563.9 [494.76–640.38]	3,389.42 [2,948.02–3,918.23]	528.33 [461.67–607.85]	−0.15 [−0.22 to −0.08]
Australasia	1,775.7 [1,587.67–1,991.45]	860.83 [766.98–968.86]	1,692 [1,502.5–1,894.93]	602.97 [528.45–682.65]	−1.59 [−1.74 to −1.45]
Caribbean	3,759.52 [3,265.35–4,263.76]	1,023.9 [895.9–1,149.68]	4,178.09 [3,600.9–4,805.63]	876.96 [755.05–1,011.48]	−0.6 [−0.69 to −0.52]
Central Asia	7,724.6 [6,807.15–8,730.75]	1,130.78 [1,004.87–1,270.77]	10,033.69 [8,702.93–11,393.91]	1,044.48 [909.34–1,184.87]	0.14 [−0.18 to 0.47]
Central Europe	30,977.29 [27,246.53–34,999.4]	2,512.34 [2,200.15–2,836.12]	20,351.62 [17,811.08–23,152.95]	1,901.41 [1,632.18–2,194.28]	−0.85 [−0.9 to −0.81]
Central Latin America	19,490.57 [16,872.57–22,881.13]	1,170.11 [1,028.29–1,331.16]	23,804.54 [20,537.54–27,381.97]	939.89 [811.76–1,080.33]	0.97 [0.33 to 1.62]
Central sub-Saharan Africa	5,244.46 [4,430.16–6,209.81]	1,063.55 [921.06–1,222.63]	11,620.96 [9774.84–13,635.69]	982.99 [844.21–1,127.36]	−0.25 [−0.35 to −0.16]
East Asia	66,252.68 [55,395.21–79,014.95]	535.26 [450.72–630.33]	165,348.28 [140,454.57–195,050.6]	981.68 [834.63–1,151.85]	2.28 [2.12 to 2.45]
Eastern Europe	80,751.81 [67,297.17–96,218.08]	3,472.2 [2,884.07–4,151.44]	54,906.65 [45,310.1–65,704.95]	2,600.36 [2,123.05–3,142.83]	−0.74 [−0.93 to −0.56]
Eastern sub-Saharan Africa	22,790.29 [19,383.17–27,061.2]	1,423.41 [1,243.35–1,630.04]	44,563.76 [37,677.55–52,692.72]	1,246.46 [1,085.85–1,425.4]	−0.53 [−0.59 to −0.47]
High-income Asia Pacific	15,449.57 [13,467.61–17,857.37]	850.09 [739.36–980.16]	8,908.91 [7,646.85–10,369.65]	473.34 [397.64–566.77]	−2.53 [−2.71 to −2.35]
High-income North America	31,454.02 [26,222.4–37,432.42]	1,101.94 [917.37–1,322.72]	33,623.21 [28,261.27–39,729.03]	904.78 [746.11–1,084.43]	−1.08 [−1.23 to −0.93]
North Africa and Middle East	31847.22 [27275.74–37739.36]	939.35 [818.92–1,077.33]	47,396.43 [40,723.74–54,593.42]	768.97 [667.85–880.58]	−0.68 [−0.71 to −0.66]
Oceania	399.74 [336.7–464.14]	684.75 [589.31–783]	859.61 [729.26–993.93]	688.42 [592.23–789.11]	−0.06 [−0.13 to 0.02]
South Asia	190,939.53 [156,205.16–230,241.28]	1,894 [1,577.63–2,252.26]	435,654.1 [356,932.82–524,541.88]	2,370.89 [1,954.52–2,843.26]	0.87 [0.47 to 1.27]
Southeast Asia	48,033.54 [40,763.7–56,349.51]	1,040.34 [898.56–1,196.41]	64,886.26 [56,005.63–74,700.39]	924.25 [800.78–1,065.94]	−0.61 [−0.69 to −0.52]
Southern Latin America	3,648.36 [3,180.23–4,137.28]	729.78 [637.51–824.49]	5,342.3 [4,589.81–6,193.28]	795.91 [681.36–923.04]	0.23 [0.18 to 0.29]
Southern sub-Saharan Africa	4,846.08 [3,955.63–5,848.59]	1,045.54 [877.96–1,244.39]	6,510.67 [5,391.37–7,818.8]	859.26 [721.53–1,029.01]	0.15 [−0.34 to 0.63]
Tropical Latin America	12,035.92 [9,918.84–14,511.47]	777.18 [654.29–920.91]	17,635.05 [14,706.66–20,862.07]	764.07 [638.5–904.25]	0.25 [−0.03 to 0.54]
Western Europe	33,159.89 [29,235.18–36,893.92]	883.04 [769.89–992.81]	20,486.47 [17,899.47–23,393.64]	522.36 [446.51–612.42]	−2.24 [−2.4 to −2.07]
Western sub-Saharan Africa	19,387.8 [16,599.37–22,443.2]	1,237.86 [1,080.4–1,413.6]	51,003.99 [43,080.08–59,410.26]	1,366.64 [1,189.24–1,563.32]	0.51 [0.42 to 0.6]

***ASIR:*** age-standardized incidence rate per 100,000 people.

**Table 2 ijerph-19-16479-t002:** Deaths, ASDR, and temporal trends for RI in 1990 and 2019.

	1990		2019		1990–2019
	Deaths Cases No. ×10^2^ (95% UI)	ASDR per 100,000 No. (95% UI)	Deaths Cases No. ×10^2^ (95% UI)	ASDR per 100,000 No.(95% UI)	EAPC No. (95% CI)
Overall	11,134.11 [10,470.07–12,096.11]	21.92 [20.65–23.86]	11,982.89 [10,600.41–13,048.31]	14.99 [13.29–16.32]	−1.29 [−1.44 to −1.14]
**Gender**					
Female	3,049.17 [2,841.52–3,274.21]	11.92 [11.16–12.71]	2,976.31 [2,709.96–3,265.94]	7.36 [6.7–8.06]	−1.73 [−1.88 to −1.58]
Male	8,084.94 [7,565.01–9,072.09]	32.23 [30.14–36.22]	9,006.59 [7,639.42–9,906.22]	22.79 [19.37–25.11]	−1.14 [−1.29 to −0.98]
**SDI region**					
High SDI	1,518.09 [1,488.69–1,546.7]	17.37 [17.04–17.7]	1,095.11 [1,017.34–1,184.18]	9.26 [8.59–10.08]	−2.47 [−2.61 to −2.33]
High-middle SDI	2,373.83 [2,247.55–2,646.26]	20.53 [19.43–22.87]	1,970.78 [1,762.01–2,152.88]	11.96 [10.77–12.98]	−1.91 [−2.21 to −1.6]
Middle SDI	4,120.39 [3,838.79–4,590.16]	26.25 [24.42–29.36]	4,380.76 [3,820.35–4,807.44]	17.45 [15.32–19.13]	−1.28 [−1.48 to −1.08]
Low-middle SDI	2,018.53 [1,835.67–2,255.58]	21.11 [19.25–23.4]	2,837.15 [2,419.48–3,162.24]	17.24 [14.69–19.14]	−0.67 [−0.79 to −0.54]
Low SDI	1,096.89 [918.07–1,274.98]	25.47 [21.84–28.65]	1,691.9 [1,403.31–2,013.27]	19.9 [16.76–23.22]	−0.85 [−0.95 to −0.76]
**GBD Region**					
Andean Latin America	81.79 [73.62–94.25]	25.42 [22.98–29.47]	115.79 [91.79–140.56]	18.75 [14.9–22.81]	−0.88 [−1.07 to −0.7]
Australasia	35.18 [34.44–35.9]	16.62 [16.28–16.98]	19.47 [18.32–20.46]	5.98 [5.63–6.29]	−3.58 [−3.73 to −3.43]
Caribbean	68.41 [64.58–80.02]	20.3 [19.2–23.9]	66.35 [53.73–84.29]	13.57 [10.98–17.25]	−1.41 [−1.6 to −1.22]
Central Asia	127.7 [122.81–131.32]	20.09 [19.3–20.65]	120.93 [108.15–134.95]	13.01 [11.66–14.5]	−0.8 [−1.27 to −0.32]
Central Europe	232.91 [228.21–236.83]	18.07 [17.7–18.39]	107 [94.45–120.43]	7.9 [6.96–8.88]	−3.24 [−3.42 to −3.07]
Central Latin America	372.98 [364.83–381.36]	27.39 [26.83–27.91]	420.7 [354.32–495.29]	16.65 [14.05–19.57]	−1.77 [−1.91 to −1.64]
Central sub-Saharan Africa	281.53 [215.46–346.69]	54.83 [42.8–70.26]	441.62 [313.2–588.97]	41.92 [30.06–58.49]	−0.92 [−1.06 to −0.78]
East Asia	2449.17 [2101.38–3338.81]	20.8 [17.87–28.52]	2,623.12 [2,172.69–3,064.37]	15 [12.56–17.34]	−0.76 [−1.21 to −0.3]
Eastern Europe	577.47 [565.32–610.37]	24.3 [23.79–25.65]	297.9 [264.95–337.88]	13.3 [11.87–14.96]	−2.48 [−2.92 to −2.03]
Eastern sub-Saharan Africa	346.88 [290.37–406.11]	26.54 [22.88–30.27]	477.12 [402.84–564.32]	18.71 [16.04–21.57]	−1.31 [−1.38 to −1.24]
High-income Asia Pacific	309.48 [296.39–319.04]	16.81 [16.11–17.36]	139.52 [127.41–157.78]	4.82 [4.49–5.67]	−5.19 [−5.48 to −4.9]
High-income North America	532.58 [525.27–540.25]	18.06 [17.82–18.31]	438.44 [418.7–450.69]	10.61 [10.09–10.91]	−1.98 [−2.19 to −1.77]
North Africa and Middle East	1,304.29 [1,055.64–1,451.78]	42.97 [33.96–47.32]	1,489.91 [1,194.03–1,747.5]	26.3 [21.04–30.75]	−1.71 [−1.73 to −1.68]
Oceania	15.05 [12.17–18.62]	27.33 [22.33–33.73]	28.14 [21.79–37.03]	23.93 [18.78–30.95]	−0.4 [−0.45 to −0.35]
South Asia	1,477.65 [1,295.62–1,639.38]	16.6 [14.45–18.38]	2,427.2 [1,904.6–2,820.79]	14.43 [11.36–16.75]	−0.5 [−0.68 to −0.32]
Southeast Asia	1,178.15 [1,032.07–1,292.54]	27.89 [24.8–30.28]	1,128 [957.71–1,324.94]	16.75 [14.27–19.7]	−1.97 [−2.08 to −1.85]
Southern Latin America	69.81 [68.07–71.49]	14.48 [14.12–14.82]	91.77 [86.97–96.36]	12.65 [11.98–13.28]	−0.49 [−0.65 to −0.34]
Southern sub-Saharan Africa	223.49 [184.66–243.76]	49.87 [41.1–54.86]	242.62 [205.36–274.05]	32 [27.6–35.81]	−1.85 [−2.25 to −1.46]
Tropical Latin America	447.43 [431.67–462.95]	32.92 [31.75–34.07]	460.3 [437.94–479.79]	19.21 [18.25–20.06]	−1.56 [−1.72 to −1.39]
Western Europe	671.01 [661.41–679.04]	15.9 [15.69–16.1]	261.47 [249.56–270.94]	4.9 [4.71–5.07]	−4.69 [−4.96 to −4.43]
Western sub-Saharan Africa	331.15 [274.4–389.7]	21.44 [18.28–24.9]	585.5 [466.55–718.11]	17.9 [14.58–21.49]	−0.53 [−0.62 to −0.44]

***ASDR:*** age-standardized deaths rate per 100,000 people.

**Table 3 ijerph-19-16479-t003:** DALYs, AS-DALYs, and temporal trends of RIs in 1990 and 2019.

	1990		2019		1990–2019
	DALYs No. ×10^2^ (95% UI)	Age-Standardized DALYs No. (95% UI)	DALYs No. ×10^2^ (95% UI)	Age-Standardized DALYs No.(95% UI)	EAPC No. (95% CI)
**Global**	712,122.4 [664,087.2–770,347.5]	1,329.47 [1,235.48–1,435.89]	729,013.26 [648,308.81–801,937.02]	917.94 [814.15–1,011.37]	−1.26 [−1.4 to −1.13]
**Gender**					
Female	208,098.7 [190,562.05–228,247.96]	780.8 [714.59–856.57]	193,677.15 [172,363.89–215,808.81]	489.05 [437.25–543.19]	−1.69 [−1.81 to −1.57]
Male	504,023.7 [467,699.93–557,320.25]	1,874.15 [1,733.5–2,062.01]	535,336.11 [463,562.97–586,867.1]	1,345.5 [1,166.1–1,474.81]	−1.09 [−1.24 to −0.94]
**SDI region**					
High SDI	88,858.35 [84,488.89–93,738.76]	1,059.2 [1,011.15–1,111.18]	60,335.63 [54,810.99–66,720.78]	562.11 [515.83–617.59]	−2.48 [−2.61 to −2.34]
High-middle SDI	150,385.15 [139,162.78–163,847.98]	1,283.56 [1,186.67–1,397.77]	120,707.72 [107,606.11–134,968.91]	763.36 [684.32–843.56]	−1.84 [−2.06 to −1.61]
Middle SDI	259579.77 [241133.39–283157.94]	1,502.58 [1,397.76–1,641.83]	248,621.98 [220,698.37–27,3031.33]	987.99 [880.75–1081.77]	−1.36 [−1.52 to −1.2]
Low-middle SDI	137,322.64 [124,626.91–152,489.3]	1,294.6 [1,171.71–1,430.64]	185,509.65 [162,145.8–207,601.08]	1,080.25 [943.53–1,209.33]	−0.59 [−0.76 to −0.43]
Low SDI	75,574.85 [62,708.46–89,015.91]	1,466.02 [1,258.01–1,680.21]	113,421.72 [95,173.88–134,008.85]	1,152.15 [977.14–1,331.52]	−0.8 [−0.9 to −0.71]
**GBD region**					
Andean Latin America	4,902.6 [4,396.78–5,555.77]	1,318.98 [1,192.05–1,505.37]	5,911.77 [4,806.04–7,089.41]	924.08 [751.11–1,106.73]	−1.09 [−1.28 to −0.9]
Australasia	2,132.23 [2,044.31–2,233.02]	1,029.38 [989.2–1,076.27]	1,156.5 [1,052.18–1,275.7]	387.26 [356.49–421.88]	−3.47 [−3.61 to −3.32]
Caribbean	4,316.62 [4,008.51–4,947.11]	1,215.42 [1,126.47–1,398.7]	3,934.24 [3,259–4,910.97]	817.37 [674.97–1,022.45]	−1.38 [−1.54 to −1.21]
Central Asia	8,090.8 [7,689.82–8,493.97]	1,206.66 [1,142.55–1,274.85]	7,489.92 [6,679.93–8,312.15]	782.8 [697.12–869.95]	−0.87 [−1.3 to −0.44]
Central Europe	16,346.65 [14,831.85–18,109.27]	1,287.56 [1,175.67–1,415.02]	8,445.05 [7,048.61–10,051.43]	648.16 [551.53–754.05]	−2.6 [−2.7 to −2.49]
Central Latin America	23,218.94 [22,345.71–24,216.2]	1,489.7 [1,425.47–1,560.82]	23,693.2 [20,326.43–27,267.39]	922.94 [792.49–1,062.49]	−1.39 [−1.54 to −1.25]
Central sub-Saharan Africa	19,360.86 [14,749.89–24,324.22]	3,085.19 [2,423.86–3,728.34]	27,415.1 [19,745.18–36,123.53]	2,190.04 [1,597.63–2,895.56]	−1.14 [−1.3 to −0.98]
East Asia	145,289.57 [126,698–192,614.2]	1,168.54 [1,020.97–1,544.6]	138,497.86 [118,706.15–157,642.08]	837.7 [723.05–944.01]	−0.93 [−1.28 to −0.57]
Eastern Europe	42,534.12 [38,443.48–47,374.99]	1,800.37 [1,645.11–1,988.66]	24,224.14 [20,793.45–28,542.97]	1,061.33 [928.45–1,226.45]	−2.04 [−2.34 to −1.73]
Eastern sub-Saharan Africa	22,892.18 [19,073.42–26,929.3]	1,355.1 [1,168.45–1,551.73]	30,358.48 [25,860.87–35,891.01]	927.02 [801.29–1,075.76]	−1.38 [−1.41 to −1.35]
High-income Asia Pacific	17,097.24 [16,178.58–18,040.11]	960.98 [911.05–1,012.71]	6,218.29 [5,538.31–7,130.11]	276.35 [249.83–319.72]	−5.16 [−5.45 to −4.87]
High-income North America	32,522.81 [30,969.3–34,315.68]	1,140.18 [1,090.86–1,196.71]	25,423.69 [23,378.27–27,639.57]	663.99 [617.66–711.86]	−2.04 [−2.22 to −1.86]
North Africa and Middle East	83,280.05 [68,073.75–94,352.54]	2,336.72 [1,919.06–2,597.23]	79,889.53 [67,371.19–92,464.84]	1,306.58 [1,096.23–1,509.14]	−1.98 [−2 to −1.95]
Oceania	929.9 [759.29–1,154.06]	1,473.91 [1,215.76–1792.4]	1,685.71 [1,335.17–2,227.61]	1,288.29 [1,025.87–1,668.58]	−0.41 [−0.46 to −0.36]
South Asia	109,861.14 [97,282.68–122,275.73]	1,129.31 [990.19–1,267.57]	182,324.75 [150,456.67–211,791.51]	1,043.94 [863.88–1,219.94]	−0.25 [−0.52 to 0.02]
Southeast Asia	75,747.96 [65,807.14–84,504.62]	1,610.09 [1,417.2–1,782.49]	64,509.58 [56,424.98–73,794.92]	928.71 [812.9–1,061.73]	−2.15 [−2.27 to −2.03]
Southern Latin America	3,997.18 [3,812.6–4,206.72]	812.95 [775.2–856.68]	5,014.58 [4,678.14–5,364.26]	719.16 [671.78–767.72]	−0.48 [−0.62 to −0.33]
Southern sub-Saharan Africa	13,189.66 [10,969.98–14,323.93]	2,674.85 [2,219.32–2,906.18]	13,313.67 [11,190.95–15,106.98]	1,658.74 [1,414.26–1,865.44]	−1.9 [−2.29 to −1.51]
Tropical Latin America	26,201.65 [25,171.8–27,343.38]	1,737.19 [1,669.2–1,811.97]	25,168.11 [23,753.68–26,462.55]	1,064.89 [1,004.12–1,120.26]	−1.36 [−1.53 to −1.19]
Western Europe	37,593.78 [35,935.68–39,545.26]	958.21 [921.19–1,002.05]	14,339.7 [13,059.96–15,807.33]	314.57 [291.18–341.19]	−4.48 [−4.75 to −4.22]
Western sub-Saharan Africa	22,616.45 [18,889–26,903.33]	1,202.26 [1,026.31–1,388.47]	39,999.37 [32,526.35–48,363.36]	1,007.76 [839.47–1,192.36]	−0.51 [−0.55 to −0.46]

***AS-DALYs:*** age-standardized DALYs rate per 100,000 people.

## Data Availability

Not applicable.

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
