# Peer review of "Global, Regional, and National Burden of Road Injuries from 1990 to 2019"

_ijerph, 2022, doi:10.3390/ijerph192416479_

Round 1

Reviewer 1 Report

The study is original and contributes to the body of knowledge. The subject matter is presented in a comprehensive manner.

I have the following remarks:

-        -There are no research questions and no hypotheses. Some of them should be added to the text because the research aim is not sufficient.

-        -Also, there is not literature review section. Probably it is made in the "Introduction" section. It would be better to divide the original introduction to the short introduction and the literature review.

-        Literature review should be improved. Authors should provide a comparison with the previous studies to highlight the research gaps and contributions. Theoretical framework needs to be strong.

-        -I suggest dividing the paper into the following parts: Introduction, Literature review, Materials and Methods, Results and Discussion, Conclusions. The literature review is small.

-       - Please give scheme of methodology.

-        Authors should provide a comparison with the previous studies to highlight the research gaps and contributions.

-     -   It is necessary to expand the conclusion. Give numerical results.

Author Response

The text is the similar to the PFD file.

Comment 1: There are no research questions and no hypotheses. Some of them should be added to the text because the research aim is not sufficient.AND. Authors should provide a comparison with the previous studies to highlight the research gaps and contributions. Theoretical framework needs to be strong. AND. Authors should provide a comparison with the previous studies to highlight the research gaps and contributions.

Response 1: We added the research gap and research questions as the reviewers suggested in lines 83-87. The revised manuscript:‘These researches include specific cities, countries and territories, continents, or regions. Furthermore, these investigations were performed in the past and need renewal. It is also important to make comprehensive of the newest condition. We wonder if the tendency in different levels (world, regions, continents, and countries) has changed in the last three decades.'

Comment 2: Also, there is not literature review section. Probably it is made in the "Introduction" section. It would be better to divide the original introduction to the short introduction and the literature review. AND. I suggest dividing the paper into the following parts: Introduction, Literature review, Materials and Methods, Results and Discussion, Conclusions. The literature review is small.

Response 2: Thank you for your suggestion on the manuscript's structure. We received the review report and reorganized the article carefully in the order of 'Introduction, Literature review, Materials and Methods, Results and Discussion, Conclusions.' as the reviewer suggested. However, it seems a little redundant when summarizing a single part for a 'Literature review' due to much information cited in the introduction and discussion. Perhaps the introduction does not clarify the research gap and needs an independent section to illustrate the sufficient research aim. Furthermore, some other lectures aiming at investigating the global burden of related diseases [1-5] usually omit the part of 'Literature review' where detailed described in 'Introduction, Discussion, and Conclusions.'. Hence, after balancing the already written order in the discussion and the introduction, we modified the expression in Response 1 to address the comparison and research gap. Similarly, the reviewer's suggestion also reminded us could add a single part to illustrate the research gap in future research.

[1] GBD 2016 Traumatic Brain Injury and Spinal Cord Injury Collaborators. Global, regional, and national burden of traumatic brain injury and spinal cord injury, 1990-2016: a systematic analysis for the Global Burden of Disease Study 2016 [published correction appears in Lancet Neurol. 2021 Dec;20(12):e7]. Lancet Neurol. 2019;18(1):56-87. doi:10.1016/S1474-4422(18)30415-0

[2] Hong C, Liu Z, Gao L, et al. Global trends and regional differences in the burden of anxiety disorders and major depressive disorder attributed to bullying victimisation in 204 countries and territories, 1999-2019: an analysis of the Global Burden of Disease Study. Epidemiol Psychiatr Sci. 2022;31:e85. Published 2022 Nov 28. doi:10.1017/S2045796022000683

[3] GBD 2019 Antimicrobial Resistance Collaborators. Global mortality associated with 33 bacterial pathogens in 2019: a systematic analysis for the Global Burden of Disease Study 2019 [published online ahead of print, 2022 Nov 18]. Lancet. 2022;S0140-6736(22)02185-7. doi:10.1016/S0140-6736(22)02185-7

[4] Malamardi S, Lambert KA, Praveena AS, Anand MP, Erbas B. Time Trends of Greenspaces, Air Pollution, and Asthma Prevalence among Children and Adolescents in India. Int J Environ Res Public Health. 2022;19(22):15273. Published 2022 Nov 18. doi:10.3390/ijerph192215273

[5] Wang Y, Li M, Guo W, Deng C, Zou G, Song J. Burden of Malaria in Sao Tome and Principe, 1990-2019: Findings from the Global Burden of Disease Study 2019. Int J Environ Res Public Health. 2022;19(22):14817. Published 2022 Nov 10. doi:10.3390/ijerph192214817

Comment 3: Please give scheme of methodology.

Response 3: Thank you for your feedback on our manuscript's writing structure. We found that we described the methodology scheme in the introduction's tail. But it seems a bit not obvious. Hence, we put the sentence' In this investigation, we used the number and rate of RI incidence, deaths, the disability-adjusted life years (DALYs), years lived with disability (YLDs), YLLs from 1990 to 2019 among men and women, social-demographic index (SDI) regions, and GBD regions, the percentage of the five RI subtypes was presented for each.' in the method section(Line 99-103) to make the scheme clear.

Comment 4:  It is necessary to expand the conclusion. Give numerical results.

Response 4: we add some notable results in conclusion. The revised part is in Lines 506-521(in red).

'RIs remain a concern in many countries and territories. Globally, the incidence decreased during 2010-2015; after which the incidence increased to 103.2 million [86.9-121.3] in 2019. Low-middle SDI countries show higher incidence, 31.1 [25.8-37.0] million in 2019; at the same time, low SDI countries have more RI related age-stanardized deaths(19.9 [16.76-23.22]) and DALYs rates(1152.15 [977.14-1331.52]). The higher DALYs rate increases burden on low SDI regions, affecting economic development. In the high SDI region, RI incidence(-1.37 [-1.47 to -1.27]), death rate(-2.47 [-2.61 to -2.33]), and DALYs(-2.48 [-2.61 to -2.34]) decrease obviously with higher EAPC values by than other regions.'

Reviewer 2 Report

This study presents the burden of road injuries from 1990-2019. The data have been obtained from The Global Burden of Diseases Study (GBD) database. Given the recent global discussions on road safety issues, e.g., the Decade of Action for Road Safety, and World Day of Remembrance for Road Traffic Victims, the topic is important and timely. Overall, the paper is well written and the data/ results are nicely presented. My main comments on this manuscript are as follows.

The long name for "RI" (line 18 in the abstract) should be given. And, please check if the long names for other abbreviations are also given.

Line 30 in the abstract - What kinds of "burden on public" health have resulted? It is better to provide explanations in the Discussion and Conclusions sections.  

In the Results, Discussion and Conclusions sections, I can see the authors have referred to “high-income countries” (Figures 2,3, Table 1 and related explanations). Furthermore, I can see that the authors mentioned: “Countries with more motor vehicles, such as Australia…” (lies 500-501). How or from where the data for income levels and motor vehicle ownership or use were obtained? Were these data obtained from other databases OR were they available in the same GBD database? Some details are needed to be mentioned. 

The authors can also provide some discussions on the potential measures that can be implemented to reduce the burdens. 

Author Response

The text is the similar to the PFD file.

Comment 1: The long name for "RI" (line 18 in the abstract) should be given. And, please check if the long names for other abbreviations are also given.

Response 1: Thank you for the constructive advice about the details in the manuscript. Then we spelled out the full name before abbreviations. Line 17 and Lines 25-26 have been changed in red.

'(1) Background: Understanding occurrence can help formulate effective preventative laws and regulations. However, the most recent global burden and road injuries (RIs) trends have not been reported. This study reports the burden of RIs globally from 1990 to 2019.'

'Age-standardized incident rate (ASIR) was highest in low-middle SDI regions, age-standardized deaths rates (ASDR) was high in middle SDI regions, and Age-standardized DALYs increased in low SDI regions.'

Comment 2: Line 30 in the abstract - What kinds of "burden on public" health have resulted? It is better to provide explanations in the Discussion and Conclusions sections.  

Response 2: We add further discussion of the 'burden on the public in the discussion and conclusion.As other reviewers also point out this issue. We only figure out the burden in the introduction(Lines 53-81) but none in the discussion. We recognized the fault and reorganized the discussion. Lines 414-436

'The results suggest low-middle SDI countries are in an awkward position; on the one hand, they need to build more roads to promote development, on the other hand, accident burden increases hampering economic growth. Though their death rate and DALYs rate are even lower than the condition in the low SDI region, low-middle SDI (such as India)  countries have the highest number and age-normalization rate calculated by incidence. In India, the RI incidence rate is not on low, owing to India having the second-largest population in the world. It is difficulties for most low-middle SDI countries; though the speed of increase of RI incidence is lower (ASIR-EAPC), the rate of decreased deaths (ASDR-EAPC) and DALYs (AS-DALYs-EAPC) remains lower than in the other four SDI regions. This phenomenon appears obviously in South Asian countries. Low-SDI countries have the highest ASDR and AS-DALYs compared to the low-middle SDI countries. We should consider that the medical level in those low SDI regions is still a public health problem which is obvious in African countries. Hence in most low SDI countries the burden of RIs is still huge.Due to their economic development and road conditions, the overall speed of various vehicles will be slower than those in developing faster countries, mainly middle and high-middle SDI regions (such as China and Russia). Hence when traffic accidents occur, middle SDI regions accidents are more severe, causing more deaths and DALYs than low-middle SDI countries. For some middle SDI countries, though the government can afford the economic loss of RIs, its EAPC of ASIR is still higher even than low-middle SDI. Taken together, the condition in low-middle SDI regions become cautious the prevention in RIs; for those in low-middle regions that turn to higher SDI index, low-middle SDI regions could be an example when balancing the economic and road injuries prevention.

'

Comment 3: In the Results, Discussion and Conclusions sections, I can see the authors have referred to "high-income countries" (Figures 2,3, Table 1 and related explanations). Furthermore, I can see that the authors mentioned: "Countries with more motor vehicles, such as Australia…" (lies 500-501). How or from where the data for income levels and motor vehicle ownership or use were obtained? Were these data obtained from other databases OR were they available in the same GBD database? Some details are needed to be mentioned. 

Response 3: It is our mistake in writing. In this sentence, we want to explain that the continent of Australasia, not Australia has most proportion of motor vehicle injuries. Australasia includes New Zealand and Australia. Hence we reorganized the sentence. Lines 524-526

We sincerely appreciate the reviewer's detailed audit to improve the quality of our manuscript!

‘Countries with more motor vehicles injuries incidence, such as in Australasia, will likely see an increase in the proportion of motor accident deaths and DALYs.’

Comment 4: The authors can also provide some discussions on the potential measures that can be implemented to reduce the burdens. 

Response 4: we add related specific potential measures in the manuscript (Line485-496)

‘Road injuries is a multifield issue. The use of helmets and seat belts [4], drink driving ban [5,6], speed limitation, and restriction of children's usage of transport have been proven to reduce mortality by 20-40%. Such approaches are critical to low- and middle-income nations, which account for more than 90% of annual road traffic fatalities and incidents. As previously reported, certain African countries without helmet-related laws should implement helmet-use policies to reduce RIs in motorcycle crashe.We recommend that legislation in low SDI regions and low-middle SDI regions ought to be built up around these four elements. Another solution is to improve road conditions by strengthening the road condition, such as cycle track implementation and enough street lighting. It should be highlighted that the illumination element is still debatable and will need to be verified further in future investigations.

Reviewer 3 Report

The authors examined the burden of road injuries from 1990 to 2019. Similar study had previously performed, however, finding the recent trends is worthwhile to develop effective preventive measures. Because some descriptions were obscure, please revise the following points:

1.      A previously published paper estimated the potential number of lives saved if each country implemented interventions to address risk factors for road injuries (Lancet 400. July 16, 2022). Although the authors mentioned in Conclusion as “Men over 20 years old have less RIs and DALYs with increasing age, though in women, incidence and DALYs rate remains high with increasing age” or “we recommend that administrators pay more attention to issues and regulations to protect pedestrian safety”, these descriptions were not enough. The authors have to describe the effective preventive measures for reducing the road injuries according to the present results.

2.      Some abbreviations were used without spelling out (RI, ASDR in abstract, ASDR in the body of the manuscript). Spell out before abbreviation.

3.      Although the authors showed the five RI subtypes in Line 87-88, this description should move to Line 47 because five subtypes were firstly mentioned.

4.      Please add the references following the description in Line 79-80: “Previous studies only demonstrated the RI condition in a small scale of regions and lack of the latest condition globally”.

5.      What is the “GBS” shown in Line 86?

6.      Numerical values throughout the manuscript were hard to understand. Put comma in every 3 digits for all values.

7.      For Tables 1-3, there were some clerical errors in second row, such as No. 102, 95% UI.

8.      Although the authors mentioned as “At the global level, the DALYs attributable to RIs decreased from 71212240 [66408720-77034750] in 1990 to 72901326 [64830881-80193702] in 2019 (Table 3, Figure 1C)in Line 237-238, the figures were increased.

9.      Regarding the descriptions in Line 258-261, correct the sentences as follows:  Nationally (Supplementary materials, Supplementary Figure S3), in 1990 the number of DALYs ranged from 16 [12-22] in Tokelau to 13739515 [11907510-18430633] in China. However, in 2019 DALYs increased from 10 [7-14] in Tokelau to 15593133 [12585561-18249333] in India.

10.  Delete the unnecessary space before “effective” in Line 375.

11.  Regarding the descriptions in Line 385-387, “A potential explanation for this may be that with increasing age, the physical condition of men declines more rapidly than 386 in women”, the reviewer also thinks that driving behaviors are more aggressive in men than in women.

12.  Descriptions of some references were inadequate: Lacking author name in Ref 1; lacking volume and page in Ref 8; lacking page in Ref 17; Bmj should be Brit Med J in Ref 32; Lacking page in Ref 42.

Author Response

The text is the similar to the PFD file.

Comment 1: A previously published paper estimated the potential number of lives saved if each country implemented interventions to address risk factors for road injuries (Lancet 400. July 16, 2022). Although the authors mentioned in conclusion as "Men over 20 years old have less RIs and DALYs with increasing age, though in women, incidence and DALYs rate remains high with increasing age" or "we recommend that administrators pay more attention to issues and regulations to protect pedestrian safety", these descriptions were not enough. The authors have to describe the effective preventive measures for reducing the road injuries according to the present results.

Response 1: Thank you for your constructive suggestion. We look for precious studies about the measures to alleviate the burden and combine them with the condition of our results. We add effective measures in the discussion.Lines485-496

‘Road injuries is a multifield issue. The use of helmets and seat belts [4], drink driving ban, speed limitation, and restriction of children's usage of transport have been proven to reduce mortality by 20-40%. Such approaches are critical to low- and middle-income nations, which account for more than 90% of annual road traffic fatalities and incidents. As previously reported, certain African countries without helmet-related laws should implement helmet-use policies to reduce RIs in motorcycle crashes [4].We recommend that legislation in low SDI regions and low-middle SDI regions ought to be built up around these four elements. Another solution is to improve road conditions by strengthening the road condition, such as cycle track implementation and enough street lighting. It should be highlighted that the illumination element is still debatable and will need to be verified further in future investigations.

Comment 2: Some abbreviations were used without spelling out (RI, ASDR in abstract, ASDR in the body of the manuscript). Spell out before abbreviation.

Response 2: We spelled out the full name as the reviewer advised. Line 17 and Lines 25-26 have been changed in red.

‘(1) Background: Understanding occurrence can help formulate effective preventative laws and regulations. However, the most recent global burden and road injuries (RIs) trends have not been reported. This study reports the burden of RIs globally from 1990 to 2019.’

‘Age-standardized incident rate (ASIR) was highest in low-middle SDI regions, age-standardized deaths rates (ASDR) was high in middle SDI regions, and Age-standardized DALYs increased in low SDI regions.’

Comment 3: Although the authors showed the five RI subtypes in Line 87-88, this description should move to Line 47 because five subtypes were firstly mentioned.

Response 3: Thank you for your suggestions on our manuscript's writing structure. After reviewers reminded us, we moved the subtype of RIs to Line 47. (now in Line 47). Your recommendation is thoughtful, it seems more fluent now.

Comment 4: Please add the references following the description in Line 79-80: "Previous studies only demonstrated the RI condition in a small scale of regions and lack of the latest condition globally".

Response 4: Sorry for the mistake in citing. We have added the reference already. Lines 82-83

‘Previous studies only demonstrated the RI condition in a small scale of regions and lack of the latest condition globally.’

Comment 5: What is the "GBS" shown in Line 86?

Response 5: Thank you for correcting the spelling errors in the manuscript. Sorry for our careless spelling, the word 'GBS' has been corrected to 'GBD'.Line94

Comment 6: Numerical values throughout the manuscript were hard to understand. Put comma in every 3 digits for all values.

Response 6: Sorry for our informal writing. We add comma in every three digits for all values in the main text and three tables. (The change are not marked in red )

Comment 7: For Tables 1-3, there were some clerical errors in second row, such as No. 102, 95% UI.

Response 7: The errors have been changed in three tables. Page7; Page9 ;Page11

Comment 8: Although the authors mentioned as "At the global level, the DALYs attributable to RIs decreased from 71212240 [66408720-77034750] in 1990 to 72901326 [64830881-80193702] in 2019 (Table 3, Figure 1C)" in Line 237-238, the figures were increased.

Response 8: we recheck the original data; it is 71212240 [66408720-77034750] in 1990  increased to 72901326 [64830881-80193702]in 2019. We have changed the mistake. Line 243

Comment 9: Regarding the descriptions in Line 258-261, correct the sentences as follows:  Nationally (Supplementary materials, Supplementary Figure S3), in 1990 the number of DALYs ranged from 16 [12-22] in Tokelau to 13739515 [11907510-18430633] in China. However, in 2019 DALYs increased from 10 [7-14] in Tokelau to 15593133 [12585561-18249333] in India.

Response 9: We rewrite the sentence. It is maybe clear now. Lines 264-268

‘Nationally (Supplementary materials, Supplementary Figure S3), Tokelau has the lowest DALYs whether in 1990, 16 [12-22] or 2019, 10 [7-14]. In 1990, China has the highest DALYs by 13,739,515 [11,907,510-18,430,633]. However, the condition changed slightly; after three decades, India replaced China with an amount of 15,593,133 [12,585,561-18,249,333].’

Comment 10: Delete the unnecessary space before "effective" in Line 375.

Response 10: Thank you for your reminder, the extra spaces have been deleted. Lines 382

‘. Moreover, this suggests that effective measures should be implemented to prevent road accidents.’

Comment 11: Regarding the descriptions in Line 385-387, "A potential explanation for this may be that with increasing age, the physical condition of men declines more rapidly than 386 in women", the reviewer also thinks that driving behaviors are more aggressive in men than in women.

Response 11: Thank you for your constructive comments. At present, we only aim to make sense of the cause of DALYs. However, after the reviewer’s suggestion, we found that before trying to explain the difference between DALYs, we need to explain the original number in the incidence. Hence, we add some possible reasons in the manuscript, including the reasons proposed by the reviewer. Lines 394-397

‘Also, some economic and social factors should be considered: exposure level restricted by regional legislations and less occupational-related mobility. It is also worth noting that a higher amount of driving-related worry and a more robust level of self-control have been postulated to explain women's underexposure to driving.’

Comment 12: Descriptions of some references were inadequate: Lacking author name in Ref 1; lacking volume and page in Ref 8; lacking page in Ref 17; Bmj should be Brit Med J in Ref 32; Lacking page in Ref 42.

Response 12: Ref 1 was written by the Lancet's editors. There is no author in the article, and we delete the author name in Endnote. We added another reference reminded by Ref1. After reading the author's guideline of MDPI, we correct the reference style. The reviewer reminded us to add the lacking page of some references, we carefully checked the original article and changed the information in Endnote.

Finally, we appreciate all reviewers for their great contribution to improving our manuscript.

Round 2

Reviewer 2 Report

The authors have addressed the reviewer's comments. The manuscript has also been improved accordingly.